

# Assessing the resiliency of surface water and groundwater systems

# under groundwater pumping

**Seung Beom Seo[1], Gnanamanikam Mahinthakumar[2], Sankarasubramanian Arumugam**

5     **[2]and Mukesh Kumar[3]**

[1]Department of Civil and Environmental Engineering, Institute of Engineering Research, Seoul

National University, Seoul, South Korea.

[2]Department of Civil, Construction, and Environmental Engineering, North Carolina State

10     University, Raleigh, North Carolina, USA.

[3]Nicholas School of Environment, Duke University, Durham, North Carolina, USA

Corresponding author: Seung Beom Seo (sbseo7@snu.ac.kr)





**Abstract:**

Since surface water and groundwater systems are fully coupled and integrated systems, increased groundwater withdrawal during drought may reduce groundwater discharges into the stream, thereby prolonging both systems' recovery from drought. To analyze watershed response
to basin-level groundwater pumping, we propose an uncertainty framework to understand the resiliency of groundwater and surface water systems using a fully-coupled hydrologic model under transient pumping. The proposed framework incorporates uncertainties in initial conditions to develop robust estimates of restoration times of both surface water and groundwater systems and quantifies how pumping impacts state variables such as soil moisture. Groundwater pumping
impacts over a watershed were also analyzed under different pumping volumes and different potential climate scenarios. Our analyses show that groundwater restoration time is more sensitive to variability in climate forcings as opposed to changes in pumping volumes. After the cessation of pumping, streamflow recovers quickly in comparison to groundwater, which has higher persistence. Pumping impacts on various hydrologic variables were also discussed. Given
that surface water and groundwater are inter-connected, optimal management of the both resources should be considered to improve the watershed resiliency under drought. Potential for developing optimal conjunctive management plans using seasonal-to-interannual climate forecasts is also discussed.





## 1. Introduction

Groundwater is an important source of water for various needs, including public supply, agriculture, and industry (Barlow and Leake, 2012; Sankarasubramanian et al., 2017). As the demand for groundwater use continues to rise with increasing population, groundwater resources

are depleting faster than they can be replenished (Gleeson et al., 2012). Unless sustainable withdrawal strategies are implemented, continued appropriation of groundwater resources can lead to streamflow depletion since groundwater and surface water are a tightly coupled interconnected system (Muller and Male, 1993; Winter et al., 1998; Sophocleous, 2002; Kumar et al., 2009b). Therefore, understanding of groundwater pumping impacts on water-budget

components is critical for conjunctive management of both surface water and groundwater resources.

Many studies have evaluated groundwater pumping impacts on streamflow based on conceptualized interaction between surface water and groundwater (Muller and Male, 1993; Konikow and Leake, 2014). Rasmussen et al. (2003) developed a semi-analytic integral solution

for the transient response of what associated with oscillatory pumping. Recent advances in computational power has also facilitated intensive simulations using extensive data sets (Kendy and Bredehoeft, 2006; Filimonova and Shtengelov, 2013; Konikow and Leake, 2014). Forementioned studies demonstrated the dependency of streamflow on aquifer properties and distance of the pumping wells in hypothetical cases without considering the complexity of

hydrogeological processes in actual watersheds. On the other hand, case studies evaluating the impacts of groundwater pumping on surface water resources in real hydrologic watershed systems have generally been performed using groundwater models without being fully coupled.





For example, MODFLOW (Harbaugh, 2005) has been popularly used to simulate impacts on

surface water resources due to groundwater pumping (Scibek et al., 2007; Zume and Tarhule,

2007; Garner et al., 2013). Studies also coupled a hydrologic model with a groundwater model to

evaluate groundwater pumping feedbacks on surface water (Kollet and Maxwell, 2008; Lin et al.,

2013; Condon and Maxwell, 2014). However, these studies examined groundwater pumping

impacts on future water availability at the regional scale by assuming hypothetical "periodic"

pumping and sparse representation of heterogeneities in aquifer and geology structure.

Regardless of the type of simulation model (i.e., a groundwater model alone, or a coupled

surface water and groundwater modeling approach), studies (Kendy and Bredehoeft, 2006; Zume

and Tarhule, 2007; Barlow and Leake, 2012; Garner et al., 2013) primarily focused on estimating

the equilibrium state – a steady state having reduced streamflow and groundwater storage  –

induced by "continuous/periodic" pumping series. Further, transient groundwater withdrawal

under a drought can cause adverse impacts such as depletions in streamflow and groundwater

storage (Barlow and Leake, 2012; Garner et al., 2013). Studies have also assessed groundwater

pumping impacts on water-budget components at the watershed scale under potential climate

change (e.g., Woolfenden and Nishikawa, 2014). However, these studies have mostly been

carried out in dry climate regions such as Arizona and California. (e.g., Leake and Pool, 2010;

Garner et al., 2013; Woolfenden and Nishikawa, 2014). Here, we focus on a humid basin from a

rainfall-runoff regime in which analyze the groundwater pumping impacts conditioned on

drought conditions.

The main objective of this study is to analyze transient pumping impacts on streamflow,

groundwater and other water budgets under observed and potential climatic conditions. To

account for feedback from groundwater pumping on all the hydrologic variables, we employed a





fully coupled hydrologic model, Penn-state Integrated Hydrologic Model (PIHM), which

simulates both surface and sub-surface hydrologic processes over prescribed triangular elements.

We primarily focused on the resiliency – the combined time needed for surface water or

groundwater system to reach equilibrium state after transient pumping and then to recover to pre-

pumping conditions after the cessation of groundwater pumping – of the watershed and also the

associated streamflow and groundwater depletions due to the transient pumping. Understanding

and quantifying transient pumping impacts at the river basin scale could be addressed by

analyzing the time series of water-budget components (e.g., soil moisture, streamflow,

groundwater) obtained under "pumping" and "no-pumping" conditions. We first show how the

restoration times of surface water and groundwater can vary by simple comparison of the two

(i.e. "pumping" and "no-pumping") time series. Since estimation of restoration times and

depletion volumes could be estimated only through modeling efforts, we argue that it is

important to consider relevant uncertainties. Hence, we proposed the uncertainty framework that

rigorously incorporates uncertainties in initial conditions and input variables (Li et al., 2009;

Sinha and Sankarasubramanian, 2013; Yossef et al., 2013). Based on the proposed uncertainty

framework, we developed the "null distribution of simulated streamflow" under "no-pumping"

and then compare the estimated streamflow under "pumping" to understand the pumping impacts

on watershed variables. We also considered the pumping impacts on watershed resiliency under

different demand scenarios and climatic conditions. The proposed uncertainty framework is

demonstrated for the Cape Fear watershed in North Carolina (NC) which is experiencing severe

shortages due to significant increase in demand over the past three decades (Singh et al., 2014).

Findings from the study are also discussed and generalized on how conjunctive management

plans need to be developed considering both changing climate and demand patterns.



## 2. Background

### 2.1 Study watershed

We considered Haw River basin in NC as the pilot watershed which makes up the

northern portion of the Upper Cape Fear River basin located in Piedmont region of NC, USA

(36°00'N, 79°30'W, Figure 1). The headwaters of Haw River run 177 km into the Jordan Lake

reservoir and the drainage area of the basin is 3,945 km$^2$. The Haw river primarily provides

freshwater for residential (cities such as Greensboro, Burlington, and Durham), industrial and

recreational uses. The climate of the Haw River basin is characterized by humid subtropical

climate receiving 1,138 mm of mean annual precipitation. Figure 1a presents land cover

classification of the Haw River basin. More than half of the entire watershed is still covered by

forest.

The land surface in the piedmont region is underlain by clay-rich, unconsolidated

material derived from in situ weathering of the underlying bedrock. This material, which

averages about 10 to 20 m in thickness, is referred to as saprolite (Heath, 1984). Aquifers of the

Piedmont region are localized, complex fractured metamorphic, igneous, and sedimentary rocks.

The rocks are covered almost everywhere by regolith, and most of groundwater is stored in the

shallow, porous regolith (Lindsey et al., 2006). Since unconfined groundwater generally occurs

both in the porous and shallow regolith (Heath, 1984), it is assumed that groundwater is stored in

unconfined aquifer across the entire watershed. The regolith contains water in pore spaces

between rock particles. The bedrock, on the other hand, does not have any significant inter-

granular porosity. The hydraulic conductivities of the regolith and the bedrock are similar and



range from about 0.001 to 1 m/day (Heath, 1984). With respect to recharge conditions, it is noted

that forested areas of the piedmont region have thick and very permeable soils overlain by a thick

layer of forest litter. As water needs in the Piedmont region increase and as commissioning of

new reservoirs is becoming increasingly difficult due to environmental impacts, it will be

necessary to make more intensive use of groundwater (Heath, 1984).

Figure 1e shows the potential pumping locations. Multiple locations of pumping wells

were hypothetically assigned for withdrawing water for public supply and irrigation purpose. A

total of 150 wells for public supply were uniformly distributed in urban areas (i.e., City of

Greensboro, Burlington, and Durham) and a total of 150 wells for irrigation were randomly

distributed in the agricultural land use areas (i.e., Planted/Cultivated classes from National Land

Cover Database 2006 (Fry et al., 2011)). Preliminary analysis indicated that random placement

of multiple wells in urban and agricultural areas did not affect the streamflow depletion unless

total number of wells pumping was changed. For the Haw River basin which is delineated by a

total of 781 triangular elements, every single groundwater pumping well was assigned to each

single triangular element as shown in Figure 1e. Thus, the amount of groundwater withdrawal for

public supply and irrigation were extracted from the assigned wells in the urban and agricultural

areas (red and yellow colored block in Figure 1e respectively).



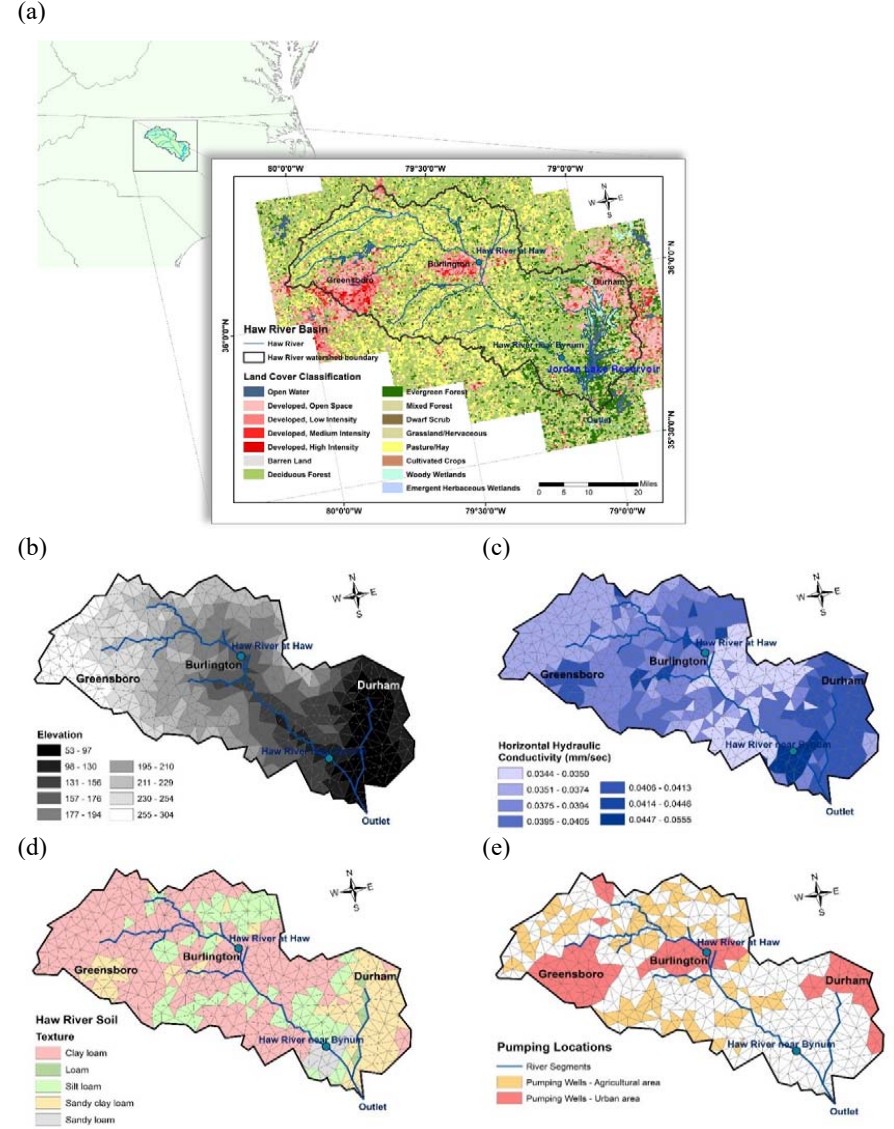

**Figure 1.** Study watershed: Haw River basin

(a) Haw River basin boundary with land cover classification map; (b) Elevation; (c) Horizontal hydraulic conductivity; (d) Soil texture classification; (e) Pumping well locations

※ (b) ~ (e) are partitioned domain maps of Haw River basin generated by PIHMgis.


## 2.2 Fully-coupled hydrologic model – PIHM

An integrated surface water and groundwater model, PIHM, was used to estimate groundwater pumping impacts on streamflow. PIHM is a fully distributed, coupled, multi-process model in which interception storage, overland flow, stream stage and groundwater states

are solved using a semi-discrete finite volume approach (Qu and Duffy, 2007). Processes simulated in the model include evaporation, transpiration, infiltration, recharge, overland flow, sub-surface flow and streamflow. For more details about the individual process equations, readers are referred to a supplement file (Text S1, Table S1-S2, and Figure S1). The model domain is discretized into unstructured grids (Kumar et al., 2009a). Laterally, hillslopes and

rivers are discretized using triangular grids and line elements, respectively. Vertically, each triangle element consists of four layers: a surface layer, a 0.25 m thick unsaturated layer, an intermediate unsaturated layer extending downward from 0.25 m to the groundwater table, and a groundwater layer. Soil moisture in the two unsaturated layers may vary from residual moisture to full saturation. Groundwater pumping in a given triangular grid is incorporated through a sink

flux term (Kumar and Duffy, 2015). PIHM has been successfully applied at multiple scales in diverse hydro-climatic regimes in both North America and Europe (e.g., Chen et al., 2015; Seo et al., 2016; Shi et al., 2013; Wang et al., 2013; Yu et al., 2013).

## 2.3 Hydroclimatic Data and Watershed Details

Daily precipitation and temperature data sets downloadable from Ed Maurer's webpage

(http://www.engr.scu.edu/~emaurer/gridded_obs/index_gridded_obs.html) were used in this study. Readers may refer to Maurer et al. (2002) for details regarding these data. These datasets are gridded observations at 1/8 degree spatial resolution and the relevant grids (total 58)





overlying the Haw River basin was selected. Daily series of precipitation and temperature for each grid were downloaded from 1951 to 2010. The data can be downloaded as either format of text or NetCDF.

Streamflow and groundwater level data for the Haw River basin were downloaded from

United States Geological Survey (USGS) Water Data webpage (http://waterdata.usgs.gov/nwis/). Haw River at Haw station (USGS HUC8 – 02096500) provides daily discharge data from 1928 to present while Haw River near Bynum station (USGS HUC8 – 02096960) has daily discharge data available from 1973 to present (see Fig. 1 for locations). Groundwater level data were available from 1948 to present as weekly series from the Chapel Hill station (USGS OR-069

355522079043001). Since the weekly variation in groundwater level is small, we converted these to monthly mean series. Figure 1b ~ 1d show partitioned domain maps of elevation, horizontal hydraulic conductivity, and soil texture classification, respectively. Each soil texture type has different values of hydraulic conductivity, porosity and vertical variation information, etc. In this study, the average size of triangular cells was approximately 5.6 km$^2$, and simulation time step

was daily. Watershed boundary, digital elevation, and land-cover/soil classification data sets for Haw River basin were obtained through the USGS national map viewer and download platform webpage (http://nationalmap.gov/viewer.html/). Land cover/soil classification data were also downloadable from the geospatial data gateway of Natural Resources Conservation Service (NRCS) webpage (http://datagateway.nrcs.usda.gov/). All spatially distributed data sets were

then converted into raster format for loading into PIHMgis (Bhatt et al., 2014), an open-source GIS-hydrologic model framework that automatically extracts hydrogeological data sets and meteorological forcings, and maps them onto model grids.



## 2.4 PIHM calibration/validation

Parameters such as horizontal/vertical hydraulic conductivities, van Genuchten drainage, and porosity were calibrated with observed groundwater levels and streamflow for the Haw River basin using daily observed precipitation and temperature time series at 1/8th degree for the

calibration period from 1956 to 1980. Because of the computationally demanding nature of the model, instead of using an automated calibration strategy that often requires search of the entire parameter space, manual adjustment was performed on hydro-geologic parameters such as hydraulic conductivity, macro-porosity, and soil retention parameters so that the values stay within physically reasonable range. The range of calibrated parameters are presented in Table 1.

Initial conditions on river segments (e.g., depth of the river) were given by observed data sets. However, initial values of sub-surface conditions were specified due to lack of data. Given the model was calibrated for 25 years after 5 years of spin-up period (1951-1955) and validated for the next 25 years, the specified initial conditions had a limited role in model performance (Table 2).

The simulated monthly streamflow and groundwater level were compared with the observed streamflow on Haw River at Haw station (USGS HUC8 – 02096500) and groundwater station at Chapel Hill (OR-069, USGS 35552209043001), respectively. Figure 2 shows comparison of monthly time series of observed and PIHM simulated streamflow and groundwater level for the calibration period (from 1956 to 1980). Correlation Coefficient (CC)

and Nash-Sutcliffe Efficiency (NSE) are compared for the calibration period and the validation period (from 1981 to 2005) in Table 2. Prediction performance of streamflow and groundwater level were reasonably good even with the manual calibration strategy employed. The variability





of groundwater level was slightly underestimated even though both streamflow and groundwater

level are simulated simultaneously in a single fully-coupled hydrologic model.

(a)

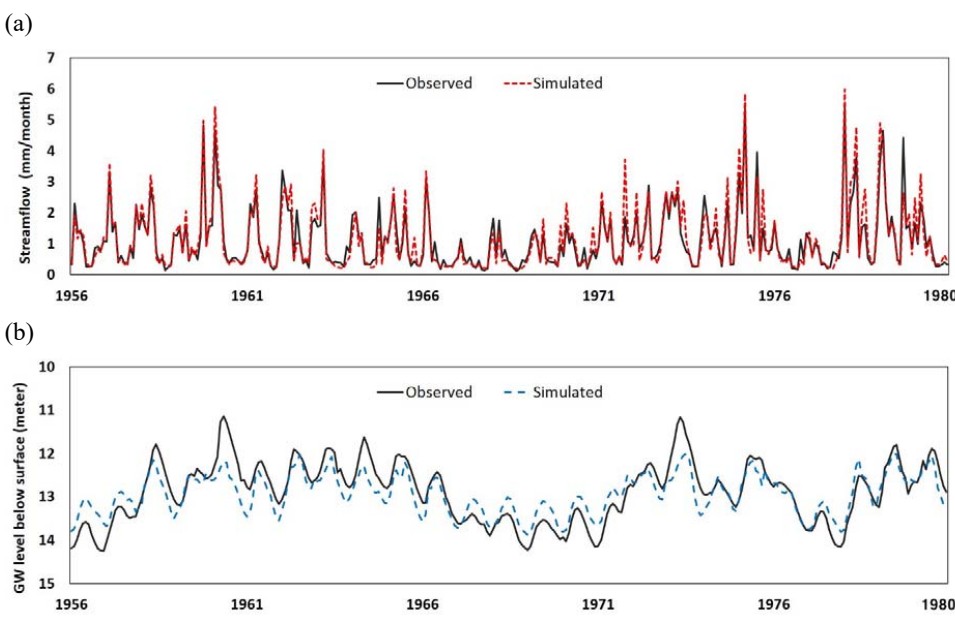

(b)

**Figure 2.** Streamflow and groundwater level series simulated by PIHM during the calibration
period (1956~1980)
    (a) Monthly streamflow series at Haw River at Haw station
    (b) Monthly groundwater level series at Chapel Hill station (without pumping)





**Table 1.** The range of key physical parameters calibrated for the Haw River basin.

| Parameter variable description | Calibrated value range | Unit |
|---|:---:|:---:|
| Vertical Saturated Hydraulic Conductivity | 1.19 ~ 1.92 | meters/day |
| Horizontal Saturated Hydraulic Conductivity | 2.97 ~ 4.79 | meters/day |
| Vertical Unsaturated Hydraulic Conductivity | 0.40 ~ 2.01 | meters/day |
| Saturated Zone Porosity | 0.420 ~ 0.434 | |
| Unsaturated Zone Porosity | 0.417 ~ 0.428 | |
| Saturated Zone Van Genuchten Soil Parameter - alpha | 0.72 ~ 1.60 | |
| Saturated Zone Van Genuchten Soil Parameter - beta | 1.73 ~ 1.84 | |
| Unsaturated Zone Van Genuchten Soil Parameter - alpha | 0.66 ~ 1.50 | |
| Unsaturated Zone Van Genuchten Soil Parameter - beta | 1.78 ~ 1.88 | |
| Vertical Macropore Hydraulic Conductivity | 30.3 ~ 150.9 | meters/day |
| Horizontal Macropore Hydraulic Conductivity | 29711 ~ 47930 | meters/day |
| Macropore Depth | 0.8 | meters |
| Maximum LAI | 0.09 ~ 7.21 | |
| Albedo | 0.14 ~ 0.28 | |
| Vegetational Fraction | 0.08 ~ 0.83 | |
| Root Zone Depth | 0.1 ~ 0.9 | meters |
| River Side Hydraulic Conductivity | 0.1 | meters/day |
| River Bed Hydraulic Conductivity | 4 | meters/day |
| River Manning's Roughness Coefficient | 4.63E-07 | |

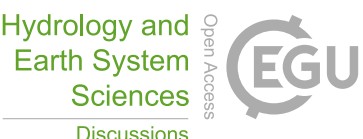

**Table 2.** Evaluation of the PIHM simulation performance for each season: Correlation Coefficient (CC) and Nash-Sutcliffe Efficiency (NSE) values of monthly streamflow and groundwater level

| Haw River basin | Station Name/Number | Calibration (1956~1980) | | | | | | | | Validation (1981~2005) | | | | | | | |
|---|---|---|---|---|---|---|---|---|---|---|---|---|---|---|---|---|---|
| | | CC | | | | NSE | | | | CC | | | | NSE | | | |
| | | DJF | MAM | JJA | SON | DJF | MAM | JJA | SON | DJF | MAM | JJA | SON | DJF | MAM | JJA | SON |
| Streamflow | at Haw USGS 02096500 | 0.95 | 0.93 | 0.94 | 0.97 | 0.84 | 0.82 | 0.86 | 0.92 | 0.94 | 0.86 | 0.88 | 0.94 | 0.77 | 0.74 | 0.73 | 0.86 |
| Groundwater Depth | USGS OR-069 355522079043001 | 0.82 | 0.80 | 0.77 | 0.77 | 0.63 | 0.59 | 0.56 | 0.57 | 0.70 | 0.69 | 0.68 | 0.67 | 0.48 | 0.47 | 0.45 | 0.45 |





## 3. Uncertainty Framework for Analyzing Water-Budget Components under Transient-Pumping

### 3.1 Experimental design

This study focused on the evaluation of transient groundwater pumping impacts on various hydrologic variables over the Haw River watershed. Figure 3 presents a conceptual experimental design of this study. We simulated hydrologic variables such as streamflow components (baseflow + overland flow), groundwater level, soil moisture depth (vadose zone thickness), evapotranspiration, recharge, and infiltration using the calibrated PIHM model. Observed daily forcing data from 1998-10 to 2010-12 were used. We defined the base scenario as 10-year simulation of these hydrologic variables from 2001 to 2010 after excluding a spin-up period preceding 27 months (2 ¼ years) from 1998-10 to 2000-12. These hydrologic variables are then simulated by the same observed daily forcings but with potential groundwater pumping scenarios. Details on the potential pumping scenarios are provided in section 3.3. Consequently, changes in the hydrologic variables due to potential pumping can be quantified by calculating the difference between the two chains – simulation without groundwater pumping (black dashed-line box in Figure 3) and simulation with groundwater pumping (red dashed-line box in Figure 3).





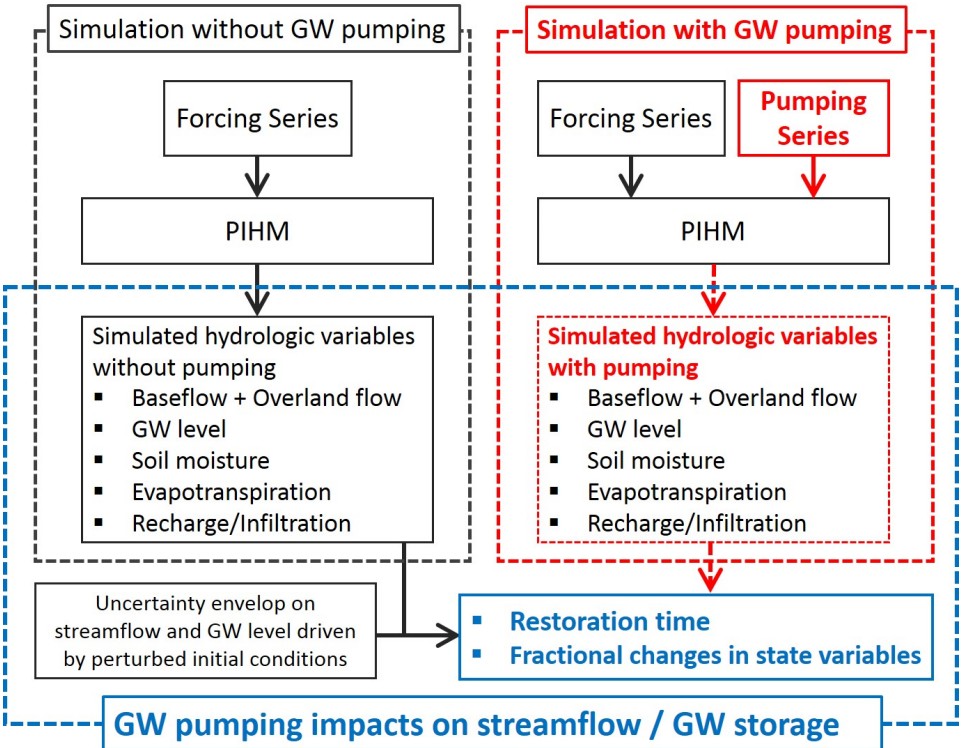

**Figure 3.** Experimental design for comparing the simulated hydrologic variables and sub-surface
storage under "pumping" and "no-pumping" conditions.

5        Apart from quantifying the differences in hydrologic variables due to pumping, we also

focused on how streamflow and groundwater storage recover back once the pumping stops,

which we call the "no-pumping state". Since potential groundwater pumping scenarios in this

study are assumed as "transient" pumping series (described in section 3.3), "no-pumping state"

mentioned above is different than the "equilibrium state" discussed in previous studies (e.g.,

10      Barlow and Leake, 2012; Garner et al., 2013; Kendy and Bredehoeft, 2006; Konikow and Leake,

2014; Leake and Pool, 2010; Scibek et al., 2007). The term "equilibrium state", as used by



Barlow and Leake (2012) and others, meets the following criteria: water levels no longer decline in response to pumping, the cone of depression does not expand any further, and the aquifer is in a new state of equilibrium in which the pumping rate is equal to the amount of streamflow depletion. The term "no-pumping state" in this study implies that streamflow and groundwater levels are restored back to the same levels as those simulated without groundwater pumping. Plus, "no-pumping state" and initial conditions (ICs) are different. ICs are initial conditions on river stage, soil moisture depth, groundwater level, etc. We define the time required for "no-pumping state" to be attained after a transient pumping as the "restoration time". For example, if restoration time of streamflow (groundwater) is estimated to be 18 months, it implies streamflow (groundwater) have returned to base scenario ("no-pumping") after 18 months. We argue that simple comparison of the time series of hydrologic variables under "pumping" and "no-pumping" could result in considerable uncertainty in the estimation of restoration time. We first show how simple comparison of the two time-series leads to inaccurate estimates of restoration time and then address that inaccuracy by developing an ensemble of hydrologic variables under "no-pumping" by perturbing the initial conditions.

### 3.2 Uncertainty envelope estimation by perturbing initial conditions

Spatial variability of initial conditions can induce uncertainties on hydrologic variables especially for short-term simulation (e.g., a year simulation or less) (Moradkhani et al., 2005) Initial conditions of land surface models also influence streamflow forecasts development (Li and Sankarasubramanian., 2012; Sinha and Sankarasubramanian, 2013; Yossef et al., 2013; Li et al., 2016). Given that a small difference in initial conditions such as soil moisture and groundwater storage could substantially alter the estimated groundwater time, we propose here an uncertainty estimation framework that perturbs the initial conditions under "no-pumping"





scenario to develop an ensemble of simulated streamflow and groundwater level. We estimated the restoration time if the time series of hydrologic variables "under pumping" recovers within this uncertainty envelope. We provide detailed steps on developing the uncertainty envelope for calculating the restoration time of hydrologic variables after pumping:

1) Simulate PIHM without pumping and obtain initial conditions from 1951 to 2010 (60 years). Here, initial conditions for 1951 such as surface water, thickness of vadose zone, groundwater level, river stage, and riverbed depth were specified and we excluded the spin-up period of first 10 years (1951-1960). Based on the remaining 50 years of simulation (1961-2010), which we call it as the base simulation, calculate the mean

annual of initial conditions, $\bar{X}_l^{Sep30^{th}}$, and standard deviation, $s_l^{Sep30^{th}}$, of initial conditions for groundwater level on Sep 30$^{\text{th}}$ which is the last date of water year, for the $l^{\text{th}}$ triangular cell. The values of calculated mean annual were then assigned as the base initial condition under pumping for each triangular cell for the simulation start date of 1998-10-1. We assumed that the initial values for each element is the climatology of those

variables.

    2) Next, perturb initial soil moisture and groundwater conditions with a noise term that follows normal distribution with zero mean and standard deviation, $s_l^{Sep30^{th}}$, for the $l^{\text{th}}$ triangular cell.

$$x_{l,m}^{Oct1^{st}} = \bar{X}_l^{Sep30^{th}} + \varepsilon_{l,m} \tag{1}$$

$$\varepsilon_{l,m} \sim N(0, w \cdot s_l^{Sep30^{th}})$$

where $x_{l,m}^{Oct1^{st}}$ is the value of $m^{\text{th}}$ perturbed initial condition for the $l^{\text{th}}$ triangular cell, $\varepsilon_{l,m}$ is random noise for $m^{\text{th}}$ perturbation of initial condition for the $l^{\text{th}}$ triangular cell, and $w$ is





variance adjustment factor ($w$: $0 < w \leq 1$). In this study, 30 sets (i.e., $m$: 1, 2,…, 30) of initial groundwater level conditions were perturbed with $w = 0.5$. Thus, one base initial conditions (mean annual for each triangular cell) and 30 sets of perturbed initial conditions of groundwater level were generated. Next, by inputting these 31 different sets

of initial conditions into PIHM, 31 sets of hydrologic variables such as streamflow and groundwater level were simulated leading to 31 series of streamflow and groundwater level for the period from 1998-10 to 2010-12. Since the analyses period of this study is from 2001 to 2010, simulation starts from 1998 to 2000 is discarded as a spin-up period.

3)    Uncertainty envelope on streamflow and groundwater level were then estimated as the

difference between the base simulation series and each perturbed series obtained with different initial conditions. Note that groundwater level is the spatial mean groundwater level over the entire watershed. We calculated spatial mean value over all elements. Difference for each series is computed as follows.

$$u_{s,m}^{t} = s_{s,m}^{t} - s_{s,base}^{t} \qquad (2)$$

$$u_{g,m}^{t} = g_{m}^{t} - g_{base}^{t} \qquad (3)$$

where $s_{s,m}^{t}$ is simulated streamflow on $s^{th}$ river segment at time $t$ driven by $m^{th}$ perturbed initial conditions, $s_{s,base}^{t}$ is simulated streamflow on $s^{th}$ river segment at time $t$ driven by the base initial conditions, $g_{m}^{t}$ is simulated groundwater level at time $t$ driven by $m^{th}$ perturbed initial conditions, $g_{base}^{t}$ is simulated groundwater level at time $t$ driven by the

base initial conditions.

Upper and lower limits of uncertainty envelope for each month were obtained from the mean monthly maximum and minimum values of the estimated envelope series, $u_{s,m}^{t}$ and $u_{g,m}^{t}$ for



streamflow and groundwater respectively, from 2001 to 2010. With the random process proposed
in this study, this is not the case as levels in some elements will be high and some will be low
and roughly evenly across the watershed, thus averaging out.

Figures 4a and 4b present mean monthly uncertainty envelope (i.e., the maximum and

minimum values of mean monthly deviances from the baseline simulation) of streamflow at the
outlet and basin-level groundwater level respectively from 2001 to 2010. Shaded areas represent
the uncertainty envelopes and dashed lines (on the secondary Y-axis) represent percent ratio of
the width of the uncertainty envelope to mean monthly streamflow and groundwater level
respectively. As shown in Figure 4a and 4b, the width of uncertainty envelope were negligible

compared to mean monthly values of baseline simulation. The width of uncertainty envelope is
in the range from 0.1 to 0.7 % for streamflow and from 0.02 to 0.06 % for groundwater level. We
next show that such small difference arising from perturbed initial conditions have a pronounced
impact on estimating the restoration time of streamflow and groundwater level after transient
pumping.

Our initial analyses on comparing the time series of hydrologic variables under
"pumping" and "no-pumping" showed large difference in the estimates of restoration time even
though both scenarios were provided with the same climate forcings (shown in Figure 5). From
Figure 5a (5b), by pumping 148 acre-ft (740 acre-ft) during the 2001 drought (Weaver 2005), we
clearly infer that groundwater depletion has almost returned to normal in 23 months (41 months)

without considering uncertainty envelope - i.e., when groundwater storage under "pumping" and
"no-pumping" becomes equal. But, the restoration time decreased to 17 months (22 months) if
the uncertainty envelope is considered - i.e., when groundwater storage under "pumping" returns
within the envelop of groundwater storage under "no pumping". The difference in restoration



times estimated using uncertainty envelope and without uncertainty envelope increased as the

pumping volume increased. Given that the initial conditions of PIHM get changed once we start

pumping, it is unreasonable to expect time series of various hydrologic variables under

"pumping" and "no-pumping" – even though both analyses are forced with the same climate

5    forcings – to have the same values after the system being fully recovered from a drought. The

developed uncertainty envelope by perturbing the initial conditions could be considered as a null

distribution against which the time series of streamflow and groundwater storage under

"pumping" has to be tested.

We also evaluated a different noise term in (1) to make it spatially correlated by

10    considering it as multivariate Gaussian distribution. However, it did not result in any changes in

the streamflow and groundwater restoration times. Since we evaluate the restoration time of

streamflow and groundwater at the outlet, impact of spatial correlation in initial conditions end

up being attenuated.


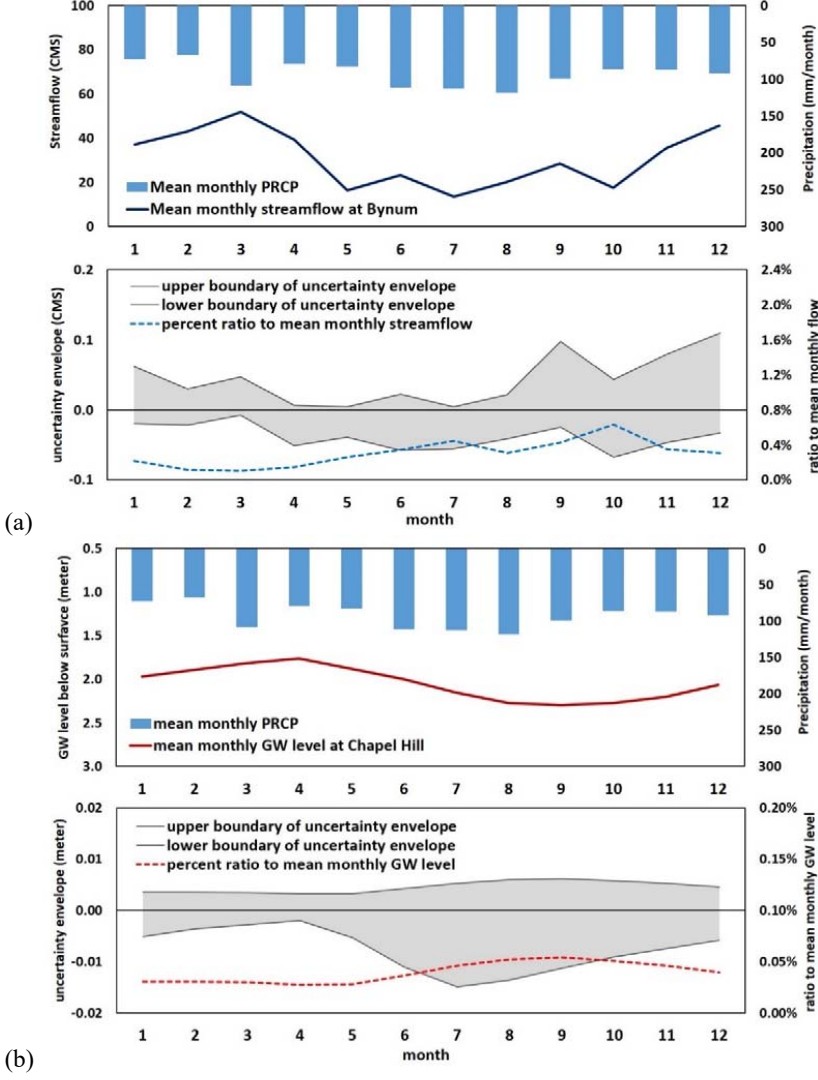

**Figure 4.** Demonstration of uncertainty envelope (differences between the simulation driven by "default" initial conditions and the 30 simulations driven by "perturbed" initial conditions) in estimating streamflow and groundwater depletion by comparing the "pumped" time series with the "no-pumped" time series, PRCP and GW are abbreviation of precipitation and groundwater, respectively.

(a) Mean monthly uncertainty envelope for streamflow at outlet - CMS: cubic meters per second

(b) Mean monthly uncertainty envelope for groundwater level





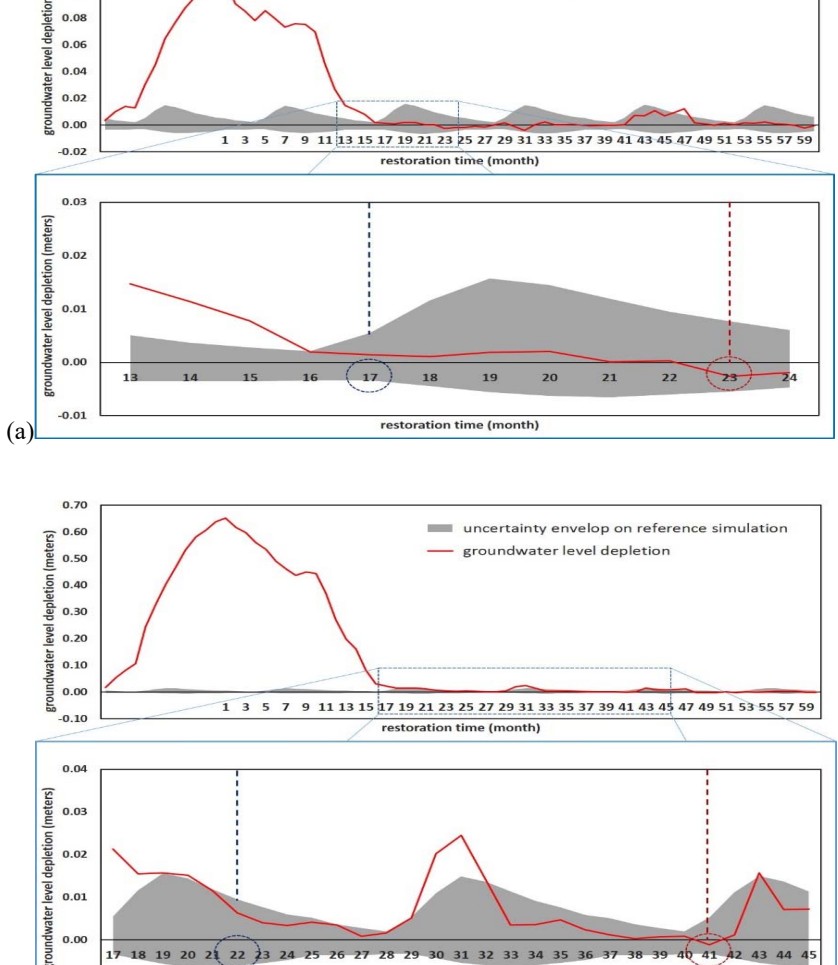

**Figure 5.** Differences in estimation of restoration time for groundwater level between "considering uncertainty envelope" driven by perturbed initial conditions (blue circle) and "no-considering uncertainty envelope" (red circle). Details on the pumping rates are described in section 3.3.

(a) 1 Below Normal Anomaly (BNA) pumping scenario – total 148 acre-feet of pumping

(b) 5 BNA pumping scenario – total 740 acre-feet of pumping



### 3.3 Potential groundwater-pumping/climate scenarios

**Potential groundwater-pumping scenarios:** To analyze the impacts of groundwater pumping on the hydrologic system, we also considered different scenarios of water withdrawals. There are limited number of wells which are monitored by government in the Haw River basin. Even for

the wells that are monitored very little pumping data is available. Due to limited availability of reliable pumping data for the reported uses, we generated potential groundwater pumping scenarios to accommodate the reduced supply during the drier conditions in the simulated streamflow. We consider the 33 percentile of the seasonal streamflow, the below-normal threshold, obtained from the simulated streamflow (1961-2010), as the reference condition to

trigger pumping. Thus,

$$wd_j = w_{BN\,j}^{thres} - \overline{w_{BN\,j}} \tag{4}$$

where $wd_j$ is water deficiency(shortfall) for $j$ season ($j$: January to March (JFM), April to June (AMJ), July to September (JAS), October to December (OND)), $w_{BN\,j}^{thres}$ is Below Normal (BN) threshold value (i.e., 33 percentile of the seasonal streamflow through the total period of

simulation) for $j$ season, and $\overline{w_{BN\,j}}$ is the mean value of all the seasonal streamflow series for $j$ season only when the seasonal streamflow of that year is lower than the value of $w_{BN\,j}^{thres}$.

Then, the sum of all the seasonal deficiencies was regarded as the total volume of the potential pumping scenario. Next, the total volume of the potential pumping was disaggregated to each season based on the current appropriation pattern for municipal and irrigation uses in the

Haw River basin. The ratio between the amount of public supply and irrigation supply in the Haw River basin is approximately 7:3 (Kenny et al., 2009). Based on this ratio, the 70 percent of



the total volume (public supply) were equally disaggregated over all the seasons, and 30 percent of the total volume (irrigation supply) were equally assigned to the two irrigation seasons, AMJ and JAS, as shown in Figure 6. And, these seasonal pumping volumes are constantly extracted across the entire season from the hypothetical pumping wells. We defined this transient seasonal

5    pumping series for one year as Below Normal Anomaly (BNA) scenario, 148 acre-feet/yr.

We considered an actual drought year 2001 for the Haw River basin as the baseline climate scenario for this study. Thus, the potential pumping scenario was applied to the year 2001 for the 10-year simulation from 2001 to 2010 to quantify the impact of pumping in depleting streamflow and groundwater. Apart from the observed shortfall in year 2001, we also

10   considered different pumping volume scenarios by multiplying each seasonal pumping volume by a prescribed scale factor. For example, 2 BNA pumping scenario can be generated by simply multiplying each seasonal pumping volume of the 1 BNA scenario by 2. For our analysis, we considered 1 BNA, 2BNA, 3BNA, 4BNA, and 5BNA.

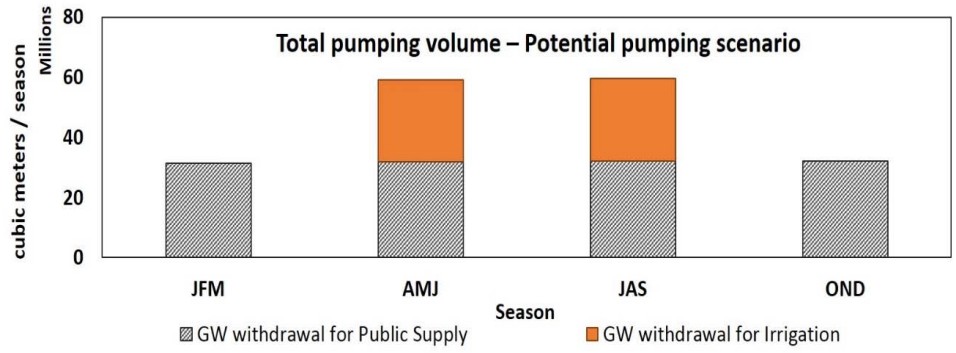

**Figure 6.** Seasonal pumping volume (m$^3$) under 1 BNA pumping scenario





**Potential climate scenarios – prolonged wet and dry climate conditions:** Apart from pumping, climate also affects hydrologic system resilience. For instance, we can expect the hydrologic system to recover faster from pumping with larger availability of water. Along with

the base climate scenario (observed climate), two potential climate scenarios were considered by replacing the first three years of forcing data set (both precipitation and temperature) from the base scenario:

- Prolonged wet scenario: forcing data from 2001 to 2003 was replaced by the historic traces of observed forcing data from 1982 to 1984 when the annual precipitation was
above normal for 3 years in a row.

- Prolonged dry scenario: forcing data from 2001 to 2003 was replaced by the historic traces of observed forcing data from 1966 to 1968 when the annual precipitation was below normal for 3 years in a row.

Figure 7 shows the annual precipitation for the three climate scenarios: the base scenario

(observed forcing), the prolonged wet scenario, and the prolonged dry scenario. The logic for considering these climate scenarios stems from the fact that the lag-1 autocorrelation on annual precipitation being close to zero, each year has equal probability of occurring. The autocorrelation is presented in Table S3 in a supplement file. Hence, we consider a low probable, but two realistic events of consecutive three wet years and three dry years to understand how

pumping would impact streamflow depletion and groundwater storage under prolonged wet and dry conditions.





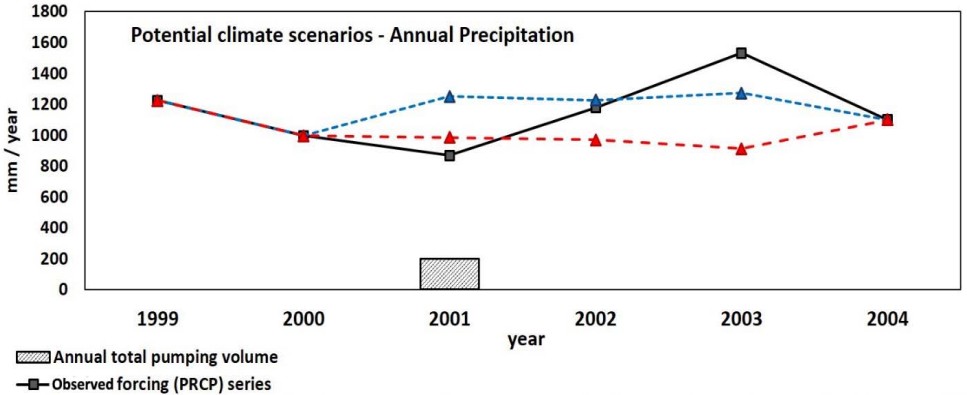

**Figure 7.** Potential wet and dry climate scenarios (dotted lines) during 2002-2004 period along with observed climate considered for the analysis. Mean annual precipitation is 1138 mm/yr.

## 4. Results

### 4.1 Groundwater pumping impacts on streamflow and groundwater depletion

BNA pumping scenarios (1 - 5 BNAs – see section 3.3 for details) were imposed on the base simulation to evaluate groundwater pumping impacts on the hydrologic system. Changes in simulated variables – overland flow, infiltration, baseflow, evapotranspiration, and groundwater storage – are calculated as the difference between "pumping" and "no-pumping" conditions. We used the uncertainty envelop for the entire analysis.

Changes in monthly time series of hydrologic variables due to pumping in 2001 are shown in Figure 8 along with observed precipitation. The differences are spatially aggregated/averaged over the entire watershed and converted into percent values to the total groundwater pumping volume, 148 acre-feet/year (183 thousand cubic meters). Groundwater





storage dropped up to approximately 70% of the total groundwater withdrawal at the end of

pumping period (2001-12) and was restored back to normal conditions after approximately 17

months (i.e., around 2003-5). Half-yearly changes in hydrologic variables and groundwater

storage from 2001 to 2003 under different groundwater pumping rates (1 BNA to 5 BNA) are

shown in Figure 9. The amount of depletions increased proportionally to the amount of pumping

volume. This means that the fractional change in each considered variable remained similar

across the changing volume of groundwater pumping. Besides, it can be inferred that

groundwater restoration time is not strictly proportional to the volume of groundwater pumping.

We found that changes in hydrologic variables due to pumping are highly dependent on

precipitation. Under wet conditions, changes in overland flow dominate in comparison to the rest

of the variables (e.g., 2002-1 and 2002-10 ~ 2002-12). This is because streamflow returned to

normal before 2013 so that there was no longer large changes in overland flow. In other words,

the overland flow under pumping is reduced compared to "no-pumping" conditions since the

pumping results in increased opportunity for infiltration (see also Figure 9a and c). On the

contrary, under dry conditions (naturally overland flow is absent), changes in baseflow and

evapotranspiration dominate indicating decreased baseflow and evapotranspiration under

pumping as compared to the "no-pumping" conditions (e.g., 2002-4 ~ 2002-9). Since 2001 was a

dry year, changes in infiltration and streamflow depletion increased significantly in 2002 which

is a wetter year than 2001. This is primarily because of depletions in groundwater storage and the

soil moisture being far below its potential under pumped conditions.

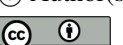



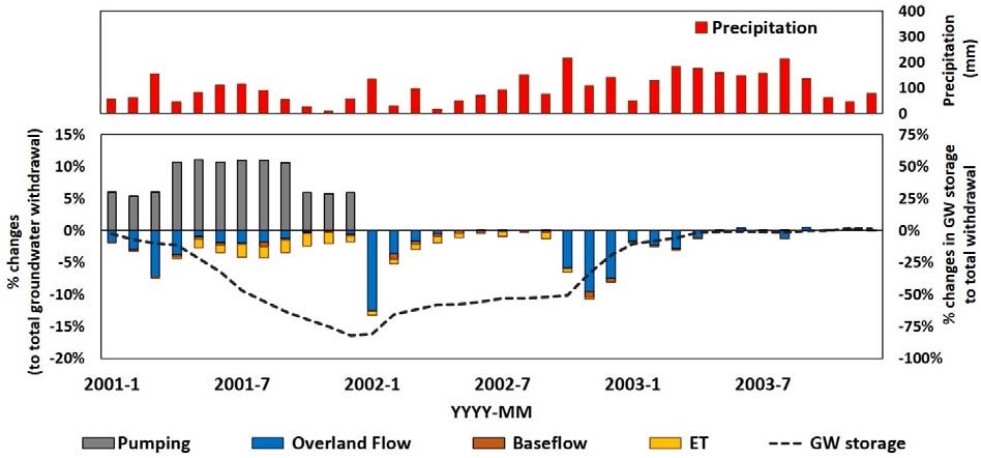

**Figure 8.** Changes in hydrologic variables between (1 BNA) "pumping" and "no-pumping" conditions under observed climate





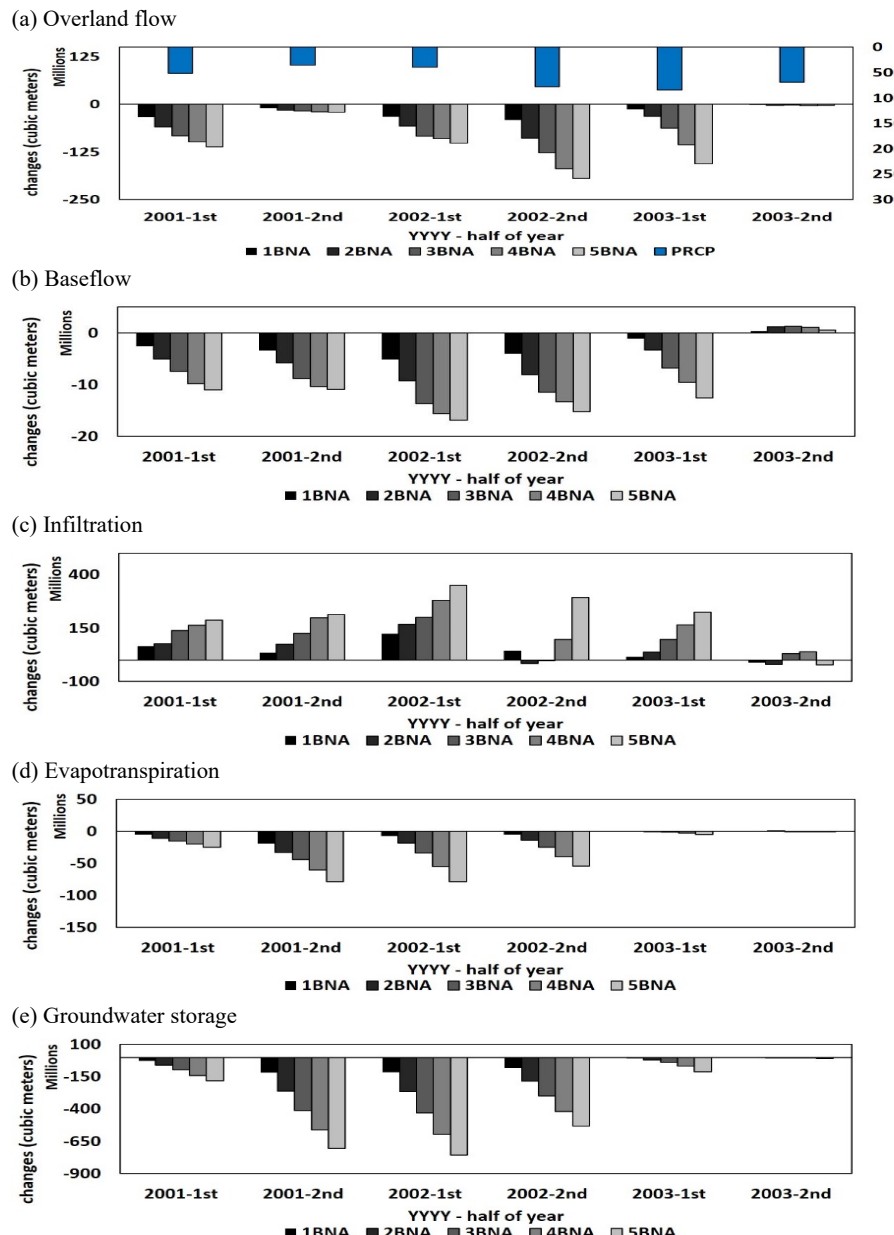

**Figure 9.** Changes in hydrologic variables between "pumping" and "no-pumping" conditions under observed climate for different pumping volume (1-5 BNA)



Table 3a shows cumulative changes in hydrologic variables such as overland flow (OF), baseflow (BF), and Evapotranspiration (ET) along with depletions in state variables such as groundwater storage (GW) and soil moisture (SM) under 1 BNA pumping scenario. Here, positive (negative) values for sub-surface variables represent the depletion (surplus) at each

5      month. As these two sub-surface variables, GW and SM, are state variables, we assume that the variables have recovered back to normal when the values become zero. We found that depletions in sub-surface water storage reached 68 % of the total pumping volume at the end of year 2001, and reached 14 %, 2 %, and 1 % at the end of year 2002, 2003, and 2004, respectively. Large depletions in groundwater storage resulted with increase in infiltration resulting in increases in

10     soil moisture (i.e., negative changes) as shown in Table 3. However, overall changes in sub-surface storage remained positive (i.e., negative changes) and gradually decreased after pumping stopped. Given that groundwater storage was nearly restored back to normal at the end of year 2003, the fraction of depletions in overland flow, baseflow, and evapotranspiration was 68 %, 10 %, and 22 % respectively to the total pumping volume. Thus, increased infiltration induced by

15     pumping results in reduction in overland flow and replenishment of groundwater storage.





**Table 3.** Changes in hydrologic variables and sub-surface storage depletion under the potential pumping scenario of 1 BNA and three different potential climate scenarios (unit: million cubic meters). BNA: Below Normal Anomaly, ET: Evapotranspiration, BF: baseflow, OF: overland flow, GW: groundwater storage, SM: soil-moisture.

(a) Observed climate

| YYYY-MM | Pumping volume | Cumulative changes in hydrologic variable | | | | Sub-surface storage depletion | | | Total change |
|---|---|---|---|---|---|---|---|---|---|
| | | ET | BF | OF | Sum | GW | SM | Sum | |
| 2001-12 | 183 | 18 | 7 | 41 | 66 (36%) | 235 | -111 | 125 (68%) | 191 (105%) |
| 2002-12 | 0 | 39 | 18 | 101 | 159 (87%) | 25 | 0 | 25 (14%) | 184 (101%) |
| 2003-12 | 0 | 40 | 19 | 119 | 178 (97%) | 0 | 4 | 4 (2%) | 181 (99%) |
| 2004-12 | 0 | 39 (21%) | 19 (10%) | 124 (68%) | 182 (99%) | 1 | 0 | 1 (1%) | 183 (100%) |

(b) Wet climate scenario

| YYYY-MM | Pumping volume | Cumulative changes in hydrologic variable | | | | Sub-surface storage depletion | | | Total change |
|---|---|---|---|---|---|---|---|---|---|
| | | ET | BF | OF | Sum | GW | SM | Sum | |
| 2001-12 | 183 | 9 | 7 | 119 | 135 (74%) | 101 | -44 | 57 (31%) | 191 (105%) |
| 2002-12 | 0 | 12 | 10 | 154 | 176 (96%) | 22 | -12 | 10 (6%) | 186 (102%) |
| 2003-12 | 0 | 12 | 9 | 158 | 179 (98%) | 7 | -6 | 1 (1%) | 181 (99%) |
| 2004-12 | 0 | 10 (5%) | 8 (4%) | 166 (91%) | 184 (101%) | -2 | 2 | 0 (0%) | 184 (101%) |

(c) Dry climate scenario

| YYYY-MM | Pumping volume | Cumulative changes on hydrologic flux | | | | Sub-surface storage depletion | | | Total change |
|---|---|---|---|---|---|---|---|---|---|
| | | ET | BF | OF | Sum | GW | SM | Sum | |
| 2001-12 | 183 | 20 | 3 | 46 | 69 (38%) | 263 | -139 | 124 (68%) | 193 (106%) |
| 2002-12 | 0 | 40 | 11 | 93 | 144 (79%) | 86 | -42 | 44 (24%) | 188 (103%) |
| 2003-12 | 0 | 45 | 15 | 107 | 166 (91%) | 37 | -20 | 17 (9%) | 183 (100%) |
| 2004-12 | 0 | 47 (26%) | 18 (10%) | 115 (63%) | 180 (99%) | 9 | -6 | 3 (2%) | 183 (100%) |



### 4.2 Groundwater pumping impacts under potential climate scenario

Along with the baseline climate scenario (observed forcing), two potential extreme climate scenarios – wet scenario and dry scenario (shown in Figure 7) – were analyzed.

Figure 10a and 10b present fractional changes in monthly overland flow, baseflow, and evapotranspiration to the total volume of groundwater pumping under wet climate scenario and dry climate scenario, respectively. Under wet climate scenario (Figure 10a), overland flow reduced substantially under pumping, which increases the opportunity for infiltration and replenishment. Thus, most depletions were captured by overland flow and diminished quickly because sufficient precipitation occurred steadily for the 3 years. On the contrary, baseflow and evapotranspiration reduces more under dry climate scenarios (Figure 10b) after pumping since groundwater storage and available soil moisture being lesser in comparison to "no-pumping" conditions. Contribution of baseflow and evapotranspiration to the total depletion increased and persisted beyond the 3 years. This is primarily due to low humidity and high temperature in summer which result in increased evapotranspiration and consequently reduced streamflow under dry climate. Besides, the changes in overland flow were also exhibited different signature between summer and winter seasons. The reductions in overland flow was greater in winter than summer season even when similar amounts of precipitation occurred (e.g., 2002-5 versus 2002-12 in Figure 10a; 2002-5 ~ 2002-6 versus 2002-12 in Figure 10b), which is primarily due to reduced soil moisture in summer resulting in limited overland flow under both "pumping" and "no-pumping" conditions. They have similar amount of precipitation, but blue colors are longer in winter 2002 than summer 2002. Thus, the changes in hydrologic variables varied with different climate conditions even though the same volume of groundwater pumping was imposed.



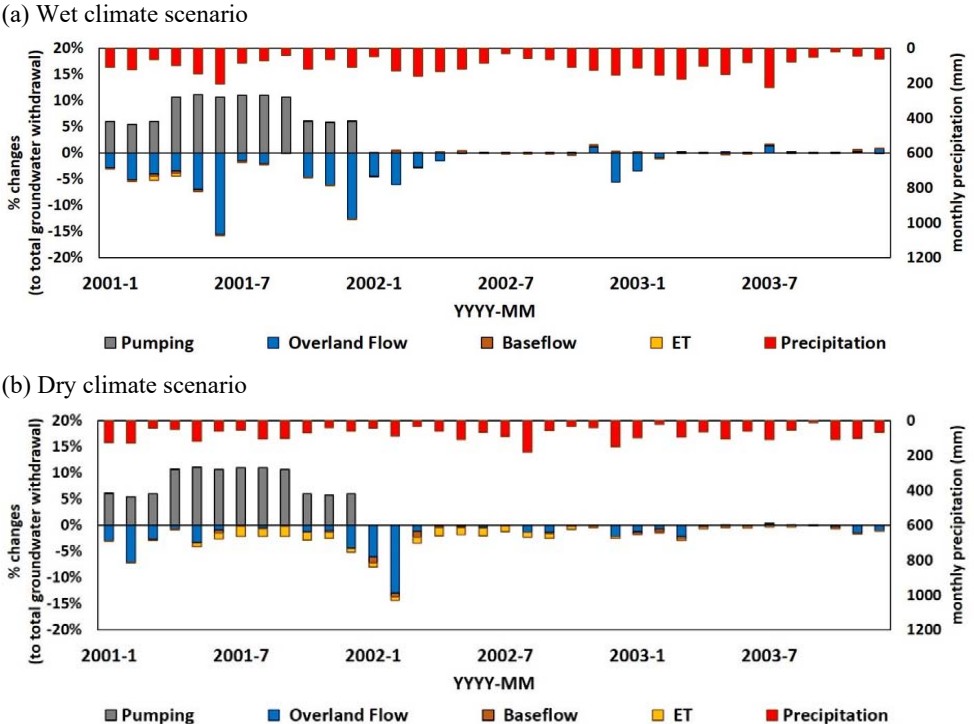

**Figure 10.** Changes in hydrologic variables between "pumping" and "no-pumping" conditions under potential climate (prolonged wet and prolonged dry) scenarios under 1 BNA pumping. ET: Evapotranspiration.
(a) Wet climate scenario
5    (b) Dry climate scenario

Table 3b and 3c show cumulative changes in hydrologic variables under wet and dry climate scenarios (observed forcing scenario was shown in Table 3a). As expected, the fractional changes to the total pumping volume, which is in parentheses, demonstrate that the mass balance

10   of the hydrologic variables is adequately captured by the PIHM model and sub-surface storage comes back to normal quickly under wet scenario but slowly under dry scenario.



### 4.3 Restoration time

Restoration time of streamflow at outlet and groundwater level under the observed climate, wet climate, and dry climate scenarios corresponding to the different pumping volumes are presented in Figure 11a and 11b, respectively. The restoration time is defined as the time required to recover to normal conditions (i.e., within the uncertainty envelope of the baseline simulation without pumping as seen in Figure 5) after the cessation of pumping.

It has been demonstrated that both streamflow and groundwater level recovered quickly (slowly) to normal conditions under wet (dry) climate in comparison to the observed climate across all pumping volumes (1 BNA to 5 BNA). The term "quickly" and "slowly" were used for relative comparison of restoration time under different climate conditions. The restoration time increased with pumping volume for all three climate scenarios. The rate of increase was more pronounced for the dry scenario, moderate for the original scenario, and nearly negligible for the wet scenario (Figure 11). Thus, prolonged dry conditions weakened the resilience of streamflow and groundwater systems due to limited water availability. Further, the relationship between restoration time and pumping volumes is non-linear while the restoration time increased monotonically for large pumping volumes. It is important to mention that over-exploitation of groundwater pumping can also impact hydro-ecosystem since it would take long time for the inflow and groundwater to come back to normal condition. This threat of over-exploitation to groundwater resources actually has been a serious issue in the regions affected by dry climate regime (e.g., Southern California and Arizona) (Draper, 2015; Lustgarten, 2015). Thus, any efforts in pumping during drier conditions should also consider potential streamflow depletion and longer restoration time of groundwater level. We intend to consider this as part of our future research.





(a)

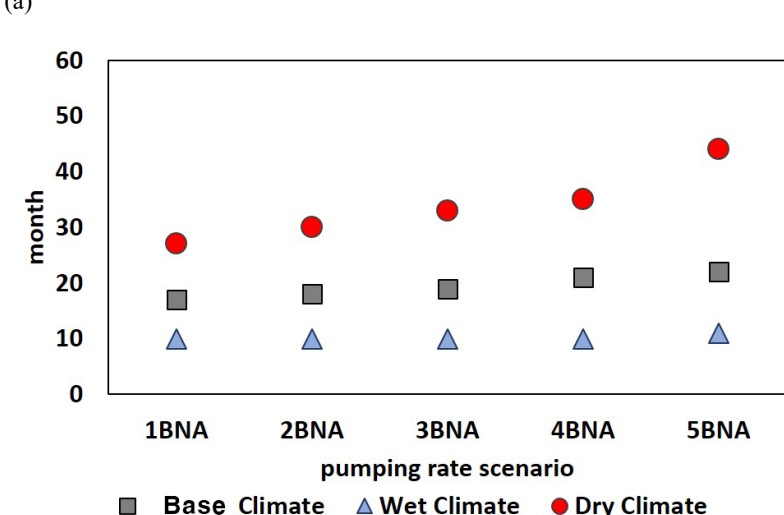

(b)

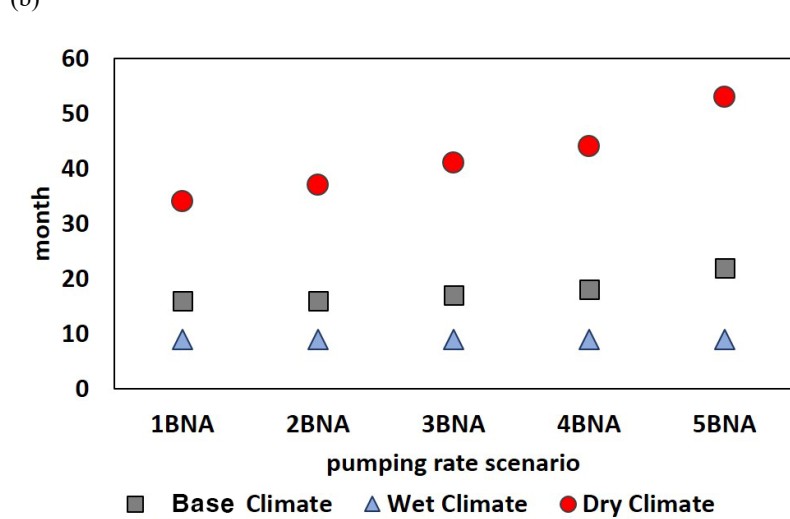

**Figure 11.** Estimated restoration time considering the uncertainty envelope framework under different pumping volume (1-5 BNA) for the considered climate scenarios
(a) Streamflow restoration time at outlet
(b) Groundwater level restoration time



## 5. Discussion and Concluding Remarks

Since transient pumping impacts at the river basin scale could be estimated only through modeling efforts, we argue that it is important to consider relevant uncertainties. Hence, we

proposed the uncertainty framework that rigorously incorporates uncertainties in initial conditions and input variables (Li et al., 2009; Sinha and Sankarasubramanian, 2013; Yossef et al., 2013). We proposed an uncertainty framework that estimates the restoration time of groundwater and surface water systems by comparing the distribution of streamflow under "no-pumping" with the streamflow obtained under "pumping". We demonstrated the application of

this framework in estimating the restoration times and the associated depletion volumes due to groundwater pumping for a humid basin in NC by considering different climatic and demand conditions. Using the uncertainty framework, the resilience of the system was examined along with the changes in water-budget components due to pumping using the integrated surface-water and groundwater model - PIHM. Pumping impacts under potential climate scenarios were

analyzed to understand how different sequences of wet and dry years could affect the restoration time and depletion volumes of streamflow and groundwater levels. We first assessed how small differences in initial watershed conditions result in significant overestimation of restoration times. Hence, instead of simply comparing two time-series of streamflow and groundwater developed under "pumping" and "no-pumping", we proposed an uncertainty framework that

perturbs initial conditions to obtain a null distribution of streamflow and groundwater under "no-pumping" for comparing the time series of water-budget components obtained under "pumping". We also suggest consideration of additional sources of uncertainty such as model, parameter and





input uncertainties for developing robust estimates of restoration time (Li and

Sankarasubramanian, 2012; Li et al., 2016).

Our analyses show that the restoration time of groundwater is more sensitive to climatic

conditions (wet vs dry) as opposed to the pumping rates. The restoration time of groundwater

and streamflow changed little with continued wet conditions. As expected, the restoration time

significantly increased with consecutive dry years. Analyses of water-budget components and

state variables revealed several critical insights. Evapotranspiration from the watershed reduced

due to low soil moisture (compared to non-pumping conditions) under pumping. Similarly,

baseflow contribution also reduced due to reduced groundwater storage under pumping over the

watershed. Since the water table is shallow (the spatial mean depth to water table was

approximately 2 meters in the model), groundwater pumping influences reductions in

evapotranspiration, baseflow and overland flow. After the cessation of pumping, increased

replenishment of groundwater storage due to enhanced infiltration also reduced overland flows.

Thus, an integrated assessment of watershed conditions due to groundwater pumping was

performed to understand how the hydrologic variables and storages of the PIHM changed due to

pumping.

This study emphasized the importance of analyzing potential pumping impacts on

watershed particularly on the ability of surface water and groundwater systems to recover during

a drought. We found significant uncertainty in recovery time exists if uncertainty in initial

conditions is not properly considered. Lack of reliable groundwater pumping data sets and

limited availability of long time series of observed groundwater data over multiple locations also

provide challenges in analyzing the impact of pumping on the watershed hydrologic conditions.

Analyses under potential climatic conditions provide pathways on how to manage the water





resources with changing climatic conditions. For instance, in the Southeast, La Nina conditions result in droughts during the winter (Devineni and Sankarasubramanian, 2010a, b). But, La Nina conditions persist for another year, this could worsen the drought conditions. However, El Nino conditions in the tropical conditions, which usually results in wet winter, would result in easing

of drought conditions (Devineni and Sankarasubramanian, 2010a, b). Thus, for future study, any efforts to manage surface water and groundwater resources during drought conditions should consider conjunctive management by quantifying the watershed resiliency conditioned on climate forecasts (Devineni and Sankarasubramanian, 2010a, b).

Analyses for the humid basin in a rainfall-runoff regime also provide insights on potential

implications for other regimes. In arid settings, it is natural to expect the competition for evapotranspiration to increase due to increased potential energy availability (Sankarasubramanian and Vogel, 2005). Hence, groundwater pumping in arid settings is further expected to increase the restoration time as the potential for evapotranspiration increases. This in turn is expected to increase the streamflow and groundwater depletion. However, in snowmelt

dominated regimes in humid basin, the opportunity for infiltration increases. Hence, we can expect relatively a quicker recovery time after pumping. These are interesting aspects that require further research to draw a synthesis on how groundwater pumping impacts various water-budget components and watershed resiliency over the Coterminous US (CONUS). We intend to consider this as part of our ongoing research in understanding the human and climate influence

watershed resiliency over the CONUS.





## Data Availability

Gridded observed precipitation and temperature data are available at Ed Maurer's webpage (http://www.engr.scu.edu/~emaurer/gridded_obs/index_gridded_obs.html). Streamflow and groundwater depth data for the target basins are available at USGS Water Data webpage (http://waterdata.usgs.gov/nwis). For watershed data sets - watershed boundary shape files, terrain data on digital elevation, land cover classification data - are available at the USGS national map viewer and download platform (http://nationalmap.gov/viewer.html).

## Acknowledgements

This research was supported in part by the National Science Foundation under grant number 1204368. Any opinions, findings, and conclusions or recommendations expressed in this material are those of the authors and do not necessarily reflect the views of the National Science Foundation.

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

model—the Ground-Water Flow Process: U.S. Geological Survey Techniques and

Methods 6- A16, variously p.

Heath, R. C. (1984), Ground-water regions of the United States (p. 78). Washington, DC: US

Government Printing Office.

Kendy, E., and Bredehoeft, J. D. (2006), Transient effects of groundwater pumping and surface-

water- irrigation returns on streamflow. *Water Resour. Res.*, 42(December 2005), 1–11.

doi:10.1029/2005WR004792

Kenny, J.F., Barber, N.L., Hutson, S.S., Linsey, K.S., Lovelace, J.K., and Maupin, M.A. (2009),

Estimated use of water in the United States in 2005: U.S. Geological Survey Circular

1344, 52 p.

Kollet, S. J., and Maxwell, R. M. (2008), Capturing the influence of groundwater dynamics on

land surface processes using an integrated, distributed watershed model. *Water Resour.

Res.*, 44(2), n/a–n/a. doi:10.1029/2007WR006004





Konikow, L. F., and Leake, S. A. (2014), Depletion and Capture: Revisiting "The Source of
Water Derived from Wells." *Groundwater*, *52*(S1), 100–111. doi:10.1111/gwat.12204

Kumar, M., Bhatt, G., and Duffy, C. J. (2009a), An efficient domain decomposition framework
for accurate representation of geodata in distributed hydrologic models. *Int. J. Geogr. Inf.
Sci.*, 23(12), 1569-1596.

Kumar, M., Duffy, C. J., and Salvage, K. M. (2009b), A Second-Order Accurate, Finite Volume–
Based, Integrated Hydrologic Modeling (FIHM) Framework for Simulation of Surface and
Subsurface Flow. *Vadose Zone J.*, 8(4), 873. doi:10.2136/vzj2009.0014

Kumar, M., and Duffy, C. J. (2015), Exploring the role of domain partitioning on efficiency of
parallel distributed hydrologic model simulations. *J. Hydrogeol. Hydrol. Eng.*, 4(1), 1-12.

Leake, S. A., and Pool, D. R. (2010), Simulated Effects of Groundwater Pumping and Artificial
Recharge on Surface-Water Resources and Riparian Vegetation in the Verde Valley Sub-
basin, Central Arizona.

Li, H., Luo, L., Wood, E. F., and Schaake, J. (2009), The role of initial conditions and forcing
uncertainties in seasonal hydrologic forecasting. *J. Geophys. Res.-Atmos.*, 114(4), 1–10.
doi:10.1029/2008JD010969

Li, W., and Sankarasubramanian A. (2012), Reducing Hydrologic Model Uncertainty in Monthly
Streamflow Predictions using Multimodel Combination, *Water Resour. Res.*, 48, W12516,
doi:10.1029/2011WR011380

Li, W., Sankarasubramanian A., Ranjithan, R.S., Sinha T. (2016), Role of multimodel
combination and data assimilation in improving streamflow prediction over multiple time
scales. *Stoch. Environ. Res. Risk Assess.*, 30: 2255. doi:10.1007/s00477-015-1158-6





Lin, H.-T., Ke, K.-Y., Tan, Y.-C., Wu, S.-C., Hsu, G., Chen, P.-C., and Fang, S.-T. (2013),
Estimating Pumping Rates and Identifying Potential Recharge Zones for Groundwater
Management in Multi-Aquifers System. *Water Resour. Manag.*, 27(9), 3293–3306.
doi:10.1007/s11269-013-0347-7

Lindsey, B. B. D., Falls, W. F., Ferrari, M. J., Zimmerman, T. M., Harned, D. A., Sadorf, E. M.,
and Chapman, M. J. (2006), Factors Affecting Occurrence and Distribution of Selected
Contaminants in Ground Water from Selected Areas in the Piedmont Aquifer System ,
Eastern United States, 1993-2003.

Lustgarten, A. (2015, July 2015), Despite decades of accepted science, California and Arizona

are still miscounting their water supplies. ProPublica. Retrieved from
https://projects.propublica.org/killing-the-colorado/story/groundwater-drought-california-
arizona-miscounting-water

Maurer, E. P., Wood, A. W., Adam, J. C., Lettenmaier, D. P., and Nijssen, B. (2002), A Long-
Term Hydrologically Based Dataset of Land Surface Fluxes and States for the

Conterminous United States. *J. Climate*, 15(22), 3237–3251.

Moradkhani, H., Hsu, K. L., Gupta, H., and Sorooshian, S. (2005), Uncertainty assessment of
hydrologic model states and parameters: Sequential data assimilation using the particle
filter. *Water Resour. Res.*, 41(5), doi:10.1029/2004WR003604

Mueller, F. A., and Male J. W. (1993), A management model for specification of groundwater

withdrawal permits. *Engineering Faculty Publications and Presentations*. 8. doi:
10.1029/92WR02908



Qu, Y., and Duffy, C. J. (2007), A semidiscrete finite volume formulation for multiprocess watershed simulation. *Water Resour. Res.*, 43(8). doi:10.1029/2006WR005752

Rasmussen, T. C., K. G. Haborak, and M. H. Young (2003), Estimating aquifer hydraulic properties using sinusoidal pumping at the Savannah River Site, South Carolina, USA, *Hydrogeol. J.*, 11, 466–482.

Sankarasubramanian A., Sabo, J.L., Larson, K.L., Seo, S.B., Sinha, T., Bhowmik, R., Vidal, A.Ruhi., Kunkel, K., Mahinthakumar, G., Berglund, E.Z. and Kominoski, J. (2017), Synthesis of Public Water Supply Use in the U.S.: Spatio-temporal Patterns and Socio-Economic Controls. *Earth's Future*. doi:10.1002/2016EF000511

Sankarasubramanian A. and Vogel, R. M. (2002), Annual Hydroclimatology of the United States, *Water Resour. Res.*, 38(6), art.no.1083, doi:10.1029/2001WR000619

Scibek, J., Allen, D. M., Cannon, A. J., and Whitfield, P. H. (2007), Groundwater–surface water interaction under scenarios of climate change using a high-resolution transient groundwater model. *J. Hydrol.*, *333*(2-4), 165–181. doi:10.1016/j.jhydrol.2006.08.005

Seo, S.B., Sinha, T., Mahinthakumar, K., Sankarasubramanian A., & Kumar, M. (2016), Identification of Dominant Source of Errors in Developing Streamflow and Groundwater Projection under Near-term Climate Change. *J. Geophys. Res. – Atmos.*, doi: 10.1002/2016JD025138

Shi, Y., Davis, K. J., Duffy, C. J., and Yu, H. (2013), Development of a Coupled Land Surface Hydrologic Model and Evaluation at a Critical Zone Observatory. *J. Hydrometeorol.*, 15, 1401–1419. doi:10.1175/JHM-D-12-0177.1





Singh, H., Sinha T. and Sankarasubramanian A. (2014), Impacts of Near-term Climate Change and Population Growth on Within-year Reservoir Systems, *J. of Water Resour. Plann. Manage.*, 140(8). doi: 10.1061/(ASCE)WR.1943-5452.0000474.

Sinha, T., and Sankarasubramanian A. (2013), Role of climate forecasts and initial conditions in developing streamflow and soil moisture forecasts in a rainfall–runoff regime. *Hydrol. Earth Syst. Sc.*, 17(2), 721–733. doi:10.5194/hess-17-721-2013

Sophocleous, M. (2002), Interactions between groundwater and surface water: the state of the science. *Hydrogeol. J.*, 10(1), 52–67. doi:10.1007/s10040-001-0170-8

Wang, R., Kumar, M., and Marks, D. (2013), Anomalous trend in soil evaporation in a semi-arid, snow-dominated watershed. *Adv. Water Resour.*, 57, 32–40. doi:10.1016/j.advwatres.2013.03.004

Weaver, J. C. (2005), The drought of 1998-2002 in North Carolina-Precipitation and hydrologic conditions, U.S. Geol. Surv. Scientific Investigations Report 2005–5053, 88 p.

Winter, T. C., J. W. Harvey, O. L. Franke, and W. M. Alley (1998), Ground water and surface water: A single resource, Circ. 1139, U.S. Geol. Surv., Denver, Colo.

Woolfenden, R., and T. Nishikawa (2014), Simulation of Groundwater and Surface-Water Resources of the Santa Rosa Plain Watershed, Sonoma County, California. U.S. Geol. Surv. Scientific Investigations Report 2014-5052, 292 p.

Yossef, N. C., Winsemius, H., Weerts, A., Van Beek, R., and Bierkens, M. F. P. (2013), Skill of a global seasonal streamflow forecasting system, relative roles of initial conditions and meteorological forcing. *Water Resour. Res.*, 49(8), 4687–4699. doi:10.1002/wrcr.20350



Yu, X., Lamačová, A., Duffy, C. J., Krám, P., Hruška, J., White, T., and Bhatt, G. (2013),
Modeling the long term water yield effects of forest management in a Norway spruces
forest. *Hydrolog. Sci. J.*, 6667(July). doi:10.1080/02626667.2014.897406

Zume, J., and Tarhule, A. (2007), Simulating the impacts of groundwater pumping on stream–
aquifer dynamics in semiarid northwestern Oklahoma, USA. *Hydrogeol. J.*, 16(4), 797–
810. doi:10.1007/s10040-007-0268-8

