# Peer review of "Assessing the resiliency of surface water and groundwater systems under groundwater pumping"

_Hydrology and Earth System Sciences, 2017_

## Referee Comment (RC1) · Anonymous Referee #1 · 5 Sep 2017

This paper presents an integrated hydrologic modeling study. Modeling techniques and approaches are standard. The model is used to assess the impact of pumping scenarios on hydrological state variables (groundwater storage, streamflow) under different climate scenarios. Special focus is on the restoration time scale, i.e. the time it takes for the system to restore from a pumping impact.

The paper includes a partial uncertainty analysis, in which model initial conditions are randomly perturbed. However, it is unclear how sources of uncertainty were identified and how it was decided which of those to include into or exclude from the analysis. Can we be sure, for instance, that uncertainty due to subsurface heterogeneity is less important that uncertainty due to unknown initial conditions?

In summary, I cannot recommend publication in HESS. I think the paper may have

value for local groundwater and surface water stakeholders in the model area, but it does not contain new knowledge, methods or insights that are of sufficient interest and importance for an international audience.

---

## Referee Comment (RC2) · Anonymous Referee #2 · 25 Sep 2017

The paper "Assessing the resiliency of surface water and groundwater systems under groundwater pumping" presents a modeling study that use PIHM to simulate the effect of groundwater pumping under different climate conditions on streamflow and groundwater storage. I think the topic fit the scope of this journal, I have several major concerns on this manuscript.

First, authors claimed their model is a "fully coupled" model. However, a single groundwater layer with a rough setting for aquifer given the size of the study area can hardly be said this is a fully coupled surface water-groundwater model. De Graaf et al. (2015) did to couple hydrological model with a groundwater model at the global scale with two layers, not even them can claim a fully coupled model. Furthermore, groundwater pumping is a key factor in this study, the oversimplified aquifer setting makes me

question the modeling results.

Second, following my first major comment, I cannot tell the purpose of this paper. Is it a methodology paper or a case study paper? I am not convinced that this is a methodology paper since due to those reasons I mentioned above. But for a case study paper, most results that presented in the manuscript are straightforward (e.g. Fig 10: wet condition will result in more runoff; Fig11: more pumping will result in long restoration time) and no further explanation or discussion are given regarding how these results might be adapted by local stakeholders/decision makers.

Third, I don't think authors clearly explain what they mean by "uncertainty framework." Is it Figure 3? Also, model uncertainty is a very population topic and can be summarized into three categories: data uncertainty, parameter uncertainty, and model structure uncertainty. Authors only test the "initial condition" (which is a very bad choice of name in my opinion) for their model and ignore all other equally (if not more) important uncertainty. I don't think their "framework" is completed since there are A LOT OF assumptions in the model setting that can affect their calculation. See my detailed comments.

Fourth, I have not convinced the concept of resiliency is well defined. Resilience is for sure the most population topic in the US but authors only use the restoration time to define resilience. This definition completely ignores the magnitude of the distortion. I suggest authors did a more comprehensive literature review on resilience index.

Given all these major concerns, I suggest editor reject this paper.

Some specific comments are given below:

P2 Line 11-12: This is only true for your case study area? I don't think you are claiming that this is a universal thing.

P2 Line 14 -15: I don't see the logic of this sentence.

P4 Line 5 - 7: I think the logic does not flow here. You mentioned no fully coupled studies before (which is completely wrong!), but are these previous studies fully coupled?

P4 Line 18-20: So this is a case study paper?

P4 Line 21: You need a summary of research gap before you going to the objective statement. You need to explain what you do in this paper. No fully coupled study before and you are the first one? Or previous fully coupled studies are not good and you are better in what way? Or previous studies have a bad assumption on pumping rare and you are better? Or why a humid basin is needed for this study?

P5 Line 13-14: What about other uncertainties I mentioned? How do you know these two are the biggest uncertainties? Are these two also the biggest uncertainties for other cases?

P6 Line 4-13: What is the percentage of these water uses in surface water vs groundwater?

P6, Line 18-20: This is weird. You focus on uncertainty. Do you know what is the uncertainty that caused by this assumption? Do you have any geological data? How do you know this assumption won't be critical? Single groundwater layer is an oversimplification for any aquifer unless you have geological data to prove it?

P 6 LIne 21- P7 Line 5: Not a single local geological information has been provided in this draft? It is really difficult for a reader to believe that you are building a groundwater model and focus on groundwater pumping?

P7 Line 6-7: Another assumption here. Again, what is the effect of this assumption on your uncertainty calculation?

P7 Line 11-13: I don't think this is a good way to convince your reader. What do you mean by not affect? How do you know? What kind of preliminary analysis? How's that affect your "uncertainty?" If there is a space issue for the main text, you can at least use supplementary materials to support this statement.

[Figure]

P9 Line 11-13: Which layer are those wells pumped? How do you know?

P10 Line 9: I did not see this station on your Fig 1.

P11 Line 7: How you do the manual calibration with 58 cells * 19 parameters assuming your model is truly distributed? If you just have one value per parameter for all 58 cells, I don't this is a distributed model.

P11 Line 15: Why not daily streamflow calibration? I believe you have daily data? And why just one surface stations, you mentioned two in the text?

P18 Line2: I don't understand what you mean by "within this uncertainty envelope?" What is "this?"

P21 Line2-4: This sentence doesn't make sense or the term "initial condition" in this paper means something else than a common definition. Why initial condition change with model runs? The initial condition is for t=0 before you even run your model. I believe you mean "Antecedent condition" not the initial condition. This is the reason I said a bad choice of name at the beginning of my comment.

P24 L8-10: What is the reasoning behind this assumption?

P27 L13: Figure 8 is not clear. I don't see "no-pumping" condition in the figure.

P28 L10: What do you mean by "dominate?" and why overland flow is dominant? The magnitude of infiltration and GW are similar if not larger than overland flow (if that's what you mean by "dominate).

P32 L3-5: Why no joint analysis with climate change and pumping scenario? Results in Figure 10 are kind of straightforward. And why not showing GW storage in Figure 10 as Figure 8?

P35 L8-10: Again straightforward result unless you are focusing on this case study and the actual restoration time means something.